# Protease and DNase Activities of a Very Stable High-Molecular-Mass Multiprotein Complex from Sea Cucumber *Eupentacta fraudatrix*

**DOI:** 10.3390/ijms23126677

**Published:** 2022-06-15

**Authors:** Anna M. Timofeeva, Irina A. Kostrikina, Pavel S. Dmitrenok, Svetlana E. Soboleva, Georgy A. Nevinsky

**Affiliations:** 1Institute of Chemical Biology and Fundamental Medicine, Siberian Division of Russian Academy of Sciences, Lavrentiev Ave. 8, 630090 Novosibirsk, Russia; bezukaf@mail.ru (A.M.T.); irina@kostrikina.ru (I.A.K.); sb543@ngs.ru (S.E.S.); 2G. B. Elyakov Pacific Institute of Bioorganic Chemistry, Far Eastern Brunch of the Russian Academy of Sciences, 159 Pr. 100 let Vladivostoku, 690022 Vladivostok, Russia; paveldmt@piboc.dvo.ru

**Keywords:** sea cucumber *Eupentacta fraudatrix*, very stable 2000 kDa protein complex, protease and DNase catalytic activities of the complex

## Abstract

Only some human organs, including the liver, are capable of very weak self-regeneration. Some marine echinoderms are very useful for studying the self-regeneration processes of organs and tissues. For example, sea cucumbers *Eupentacta fraudatrix* (holothurians) demonstrate complete restoration of all organs and the body within several weeks after their division into two parts. Therefore, these cucumbers are a prospective model for studying the general mechanisms of self-regeneration. However, there is no data available yet concerning biomolecules of holothurians, which can stimulate the processes of organ and whole-body regeneration. Investigation of these restoration mechanisms is very important for modern medicine and biology because it can help to understand which hormones, nucleic acids, proteins, enzymes, or complexes play an essential role in self-regeneration. It is possible that stable, polyfunctional, high-molecular-weight protein complexes play an essential role in these processes. It has recently been shown that sea cucumbers *Eupentacta fraudatrix* contain a very stable multiprotein complex of about 2000 kDa. The first analysis of possible enzymatic activities of a stable protein complex was carried out in this work, revealing that the complex possesses several protease and DNase activities. The complex metalloprotease is activated by several metal ions (Zn^2+^ > Mn^2+^ > Mg^2+^). The relative contribution of metalloproteases (~63.4%), serine-like protease (~30.5%), and thiol protease (~6.1%) to the total protease activity of the complex was estimated. Metal-independent proteases of the complex hydrolyze proteins at trypsin-specific sites (after Lys and Arg). The complex contains both metal-dependent and metal-independent DNases. Mg^2+^, Mn^2+^, and Co^2+^ ions were found to strongly increase the DNase activity of the complex.

## 1. Introduction

The ability to relatively quickly recover lost and damaged cells and structures of various organs is the most essential adaptation mechanism in some marine organisms [1,2,3,4,5]. Some human organs, such as the liver, are capable of very weak self-regeneration. Some marine echinoderms, including sea cucumbers, are very promising for studying the processes of self-restoration and regeneration of organs and tissues. Many marine echinoderms can achieve relatively rapid self-restoration of lost body parts following a wide variety of injuries [1,2,3,4,5]. For example, many sea cucumbers can completely restore their internal organs after a powerful breach or separation into two parts; holothurians fully restore their whole body within several weeks. However, the mechanisms of their self-regeneration have not yet been studied. It remains unclear which components of these organisms (hormones, oligosaccharides, lipids, proteins, enzymes, other compounds, or their possible complexes) can trigger and support self-regeneration. Therefore, identification of cucumber components that may be important for self-regeneration processes is important for the understanding of such mechanisms.

It was shown that many biological processes are realized by different protein complexes [6]. Many cellular processes turn on the functioning of some enzymes, which often form stable or temporary protein complexes. The functioning of such complexes leads to an increase in the specificity, efficiency, and speed of metabolic pathways [7]. The association of various proteins, enzymes, nucleic acids, and other molecules can lead to the formation of polyfunctional complexes, demonstrating the expansion of their biological functions compared to their individual elements.

Human complexes of proteins associated with placental membranes were analyzed using SDS-PAGE and MALDI mass spectrometry [8]. A total of 733 common specific proteins and 34 common proteins were found in the complexes. The existence of multiprotein complexes in the soluble fraction of human milk [9], mother placenta [10,11], and sea urchin eggs [12] was recently analyzed. The exceptionally stable complexes of various proteins (~1000 ± 100 kDa) were purified from milk, placentae, and sea urchin eggs by FPLC gel filtration [9,10,11,12]. A very stable complex ~2000 kDa was recently isolated from sea cucumbers *Eupentacta fraudatrix* for the first time [13]. This complex contains ~15 major and many minor proteins with molecular weights (MWs) > 10 kDa and >21 small proteins and peptides with MWs 2.0–8.6 kDa [13]. Like complexes from human milk and placenta, as well as sea urchin eggs [9,10,11,12], this complex was effectively destroyed only in the presence of 3.0 M MgCl_2_ and, to a lesser extent, 3.0 M NaCl, whereas the best dissociation occurs in the presence of 8.0 M urea + 0.1 M EDTA+ 3.0 M NaCl [14]. It was shown that the sea cucumber complex accumulates various elements, including those contained in very low concentrations in seawater. Some studies have shown that various metal ions can significantly affect the stability of complexes of very different nature, consisting of charged, hydrophobic and hydrophilic molecules [15,16], as well as proteins and nucleic acids [17,18,19,20,21]. The data indicate that metal-dependent interactions are formed, in addition to hydrogen bonds between the complex components [14]. The formation of complexes can lead to a vast expansion of their biological properties, including their interactions with different cells, proteins, nucleic acids, etc. In connection with the expansion of the biological functions of stable complexes, it should be noted that they possess several metal-dependent and -independent catalytic activities. Unique, highly stable complexes can exist in marine organisms capable of rapid regeneration. For example, environmental and biotic stresses stimulate nitration, as well as changes in structure and function of the sea urchin major yolk protein toposome [20,21].

The complex from milk has Mg^2+^-dependent DNase and metal independent amylase activities [9], whereas the complex from sea urchin eggs possesses phosphatase activity [12]. In the case of these complexes, only these three activities were analyzed. Many possible enzymatic activities have been analyzed in the case of a stable complex from the placenta [11], which possesses nine different enzymatic activities: DNase, RNase, ATPase, phosphatase, protease, amylase, catalase, peroxidase (H_2_O_2_-dependent), and oxidoreductase (H_2_O_2_-independent) activities. With this in mind, the analysis of stable complexes with extended biological and enzymatic functions is of particular interest.

In the present study, we used different methods (gel filtration, SDS-PAGE analysis, and MALDI mass spectrometry) to analyze, for the first time, whether sea cucumber *Eupentacta fraudatrix* protein complex possesses protease and DNase catalytic activities.

## 2. Results

### 2.1. Isolation and Analysis of Sea Cucumber Protein Complex 

Ten samples of sea cucumbers *Eupentacta fraudatrix* were subjected to homogenization and used for stable complex isolation as in [13,14]. Homogenate was concentrated and subjected to FPLC gel filtration on Sepharose 4B. Sepharose 4B efficiently separates proteins with MWs of 60 kDa. A typical profile of gel filtration is shown in Appendix A. The protein complex has an MW comparable to that of Blue dextran (~2000 kDa). After re-gel filtration of the protein complex on Sepharose 4B, only one peak was revealed with the same MW ~2000 kDa (Appendix A). After re-gel filtration, this complex was used to analyze its catalytic activities.

### 2.2. Protease Activities of the Complex 

First, protease activity of the complex was measured by a known method using azocasein as a substrate [22]. Figure 1A shows pH dependence of optical density (A_436_) changes in the reaction of azocasein hydrolysis by the complex. The complex may contain different proteases differing in optimal pH values. Taking this into account, an analysis was performed to determine the type of proteases that can be included in the complex composition. 

Preincubation of the complex with specific inhibitors of different proteases: iodoacetamide (thiol-like proteases) reduced its activity by approximately 11%, ABSF (serine-like proteases) reduced activity by 48%, and EDTA (metal-dependent proteases) reduced activity by 32% (Figure 1B). These data indicate that the complex contains all three types of proteases. The inhibition of activity of EDTA was less than that of ABSF. However, the addition of ZnCl_2_ and MnCl_2_ to the reaction mixtures led to a strong increase in the activity of the complex (Figure 1B). This means that the complex contains a certain amount of metal ions necessary for the manifestation of the activity of metalloproteases, but the addition of external metal ions leads to an increase in the activity of the complex metalloproteases.

Most often, canonical proteases in different mammals are zinc-dependent [23,24] and Ca^2+^-dependent proteases [25]. It is interesting to note which metal ions most effectively activate the metalloprotease of the complex. Zinc ions activated the protease activity of the complex better than other metal ions, although only at low concentrations (≤0.5 mM) (Figure 1C). The complex showed almost the same relative activity as in the case of Zn^2+^ ions in the presence of manganese ions but at higher 2.0 mM concentration. Mg^2+^ ions also significantly increased the protease activity of the complex, especially at the optimal concentration of 1.0 mM. Ca^2+^ ions had almost no effect on the proteolytic activity of the complex, whereas Co^2+^ ions inhibited the enzyme (Figure 1C). Thus, this protease (or proteases) of the complex can be considered to be dependent on several metal ions, namely Zn^2+^, Mn^2+^, and Mg^2+^ metalloproteases. Taking into account the effect of metal ions, the relative contribution of metalloproteases (~63.4%), serine-like protease (~30.5%), and thiol protease (~6.1%) to the total protease activity of the complex can be approximately estimated.

### 2.3. Type of Metal-Independent Proteases of the Complex 

It is interesting to understand which types of serine protease (trypsin or chymotrypsin type) are included in the stable complex. To facilitate such an understanding, we analyzed the hydrolysis sites by a complex of five histones using MALDI-TOF spectrometry (in the absence of metal ions). Figure 2, Figure 3 and Figure 4 show the MALDI spectra of hydrolysis of five histones by a high-molecular-weight complex. Based on the molecular masses of the products, the hydrolysis sites were revealed, which are indicated below the spectra in Figure 2, Figure 3 and Figure 4. 

In the absence of metal ions, the complex hydrolyzed all histones only after the amino acid residues of lysine and arginine, and there was not a single hydrolysis site after the aromatic residues of these histones (in the given histone sequences, aromatic residues are marked in grey). Thus, the stable complex of sea cucumber includes serine protease of the trypsin type. All spectra of hydrolysis products were analyzed after incubation of the reaction mixtures from several minutes to 20 h. Interestingly, the rate of hydrolysis of five histones by the protein complex differed considerably. The maximum rate was observed in the hydrolysis of H1 histone (Figure 2) and decreased approximately in the following order: H1 > H2B > H2A > H4 > H3 (Figure 2, Figure 3 and Figure 4). This is most likely related to histone packing density. As previously shown [26,27,28], unlike other histones, H1 has an almost linear shape in solution, and all sites of its hydrolysis are easily accessible.

### 2.4. DNase Activities of the Complex

Usually, pH dependences of DNase activities of different organisms are characterized by a relatively sharp peak corresponding to maximal activity with respect to dependences on pH. It is known that neutral metal-dependent human DNase I has only one pH optimum at pH 7.0–7.2, and DNases of other mammals demonstrate similar pH optima [29,30]. In addition, humans and many animals contain an acidic metal-independent DNase II with pH optima close to pH 4.9–5.0 [29,30]. Figure 5A shows the dependence of DNase activity on pH of the sea cucumber complex. The complex exhibits maximum activity in a wide pH range from 5 to 7. This may indicate that this dependence is a superposition of the pH dependences of several DNases included in the complex

As shown recently [14], the stable holothurian complex contains various metal ions. The relative content of elements (µg/g of complex powder) is as follows: Ca (3200) > Zn (51.0) > Mg (44.0) > Fe (23.0) > Ni (13.0) > Cu (7.5) > Cr (4.1) > Ba (2.5) ≈ Co (2.0) ≈ Mo (2.0) > Mn (1.6) > Cd (0.7) [14]. The obtained data indicate that the holothurian complex very strongly accumulates macro- and microelements from seawater. The concentration of various elements in the holothurian complex is 8.8–1071 times higher than in seawater [14]. It has been shown that the maximal destruction of the complex occurs in the presence of 8.0 M urea, whereas the addition of EDTA strongly increases the complex dissociation. The obtained data indicate that the formation of a stable complex occurs due to hydrogen bonds and metal-dependent interactions between the complex components [13,14]. However, a portion of the metal ions can be located within the complex, forming metal-dependent electrostatic contacts between proteins. In contrast, the other portion of the metal ions can be chelated by proteins and enzymes on the surface of the complex. These surface metal ions may be important for complex metal-dependent enzymatic activities.

As shown in Figure 5B, dialysis of the complex against EDTA and the addition of EDTA to the reaction mixture greatly reduce its DNase activity, although complete suppression of the activity is not observed. This indicates that the complex contains both metal-dependent and metal-independent DNases. Figure 5C shows the effect of various metal ions (2.0 mM) on DNase activity of the complex dialyzed against EDTA. The maximum increase in activity at a given concentration of metal ions (2.0 mM) was found for Ca^2+^ and Cu^2+^ ions. However, Mg^2+^, Mn^2+^, and Co^2+^ ions also strongly increased the DNase activity of the complex (Figure 5B). However, the optimal concentrations of all metal ions turned out to be higher than 2.0 mM. The maximum DNase activity of the complex was observed in the presence of a reaction mixture of 5.0 mM MnCl_2_, CuCl_2_, MgCl_2_, and CaCl_2_ (Figure 5C). However, the activity in the presence of Mn^2+^ ions was approximately 1.4–1.5 times higher than that in the presence of other metal ions. 

Interestingly, the optimal concentrations of metal ions in the case of the protein complex of holothurians are remarkably lower (2–5 mM) than for human DNase I (10 mM) [29,30]. In addition, mammalian DNases I splits DNAs not only in the presence of MgCl_2_ but also in the presence of CaCl_2_ [29,30]. DNase I is only slightly activated by calcium ions alone, but it is ~50-fold more active in the presence of Mg^2+^ (10 mM) in combination with Ca^2+^ (2 mM) [29,30]. In contrast to the DNase I of mammals, the metal-dependent DNase of the holothurian complex is activated by some metal ions to an approximately comparable degree. Moreover, the addition of Ca^2+^ ions to the reaction mixture containing other metal ions in the optimal concentration leads to very weak activation of the complex in DNA hydrolysis: only ~5–10% (Figure 5D).

### 2.5. In Situ Assay of DNase Activity

It is interesting to understand how many DNases of holothurians are included in their stable complex. To this end, we incubated the complex with 0.1% SDS and subjected it to SDS-PAGE in a gel-containing DNA. After removal of SDS and incubation of the gel in a standard reaction mixture, hydrolysis of polymeric DNA occurred in zones corresponding to proteins with the following MWs: 279.8 ± 15.0, 35.3 ± 5.0, 23.4 ± 3.0, and 15.9 ± 2.0 kDa (Figure 6). 

There are no available data in the literature on DNases with MWs higher than 250 kDa. 

All known human and mammalian DNases generally have relatively low molecular weights, i.e., 15–35 kDa [29,30,31,32,33]. Several DNases of the sea urchin *Strongylocentrotus intermedius* have been described: Ca^2+^/Mn^2^-dependent (25 kDa; optimal pH = 8.5), Ca^2+^/Mg^2+^-dependent (~63 kDa; optimal pH = 7.5.), and acid A-DNase (37 kDa; optimal pH = 5.5) [34,35,36].

Taking this into account, it is likely that this activity in the 279.8 kDa band of holothurian complex (Figure 6) corresponds to a fragment of an incompletely dissociated complex in the presence of SDS. 

### 2.6. ATPase Activity of the Complex 

Classical ATPases are enzymes of the membranes of all animal cells; they specifically accumulate in the cell potassium ions and pump Na^+^ ions outward using ATP energy [37,38]. A specific protein complex containing ATPase has been identified; this enzyme powers lipopolysaccharide transport from the cytoplasmic membrane across the cell envelope [39]. ATPase with high activity was found in a highly stable multiprotein complex of the female placenta [11]. With this in mind, we analyzed ATPase activity in the holothurian complex. In contrast female placental complex, we did not observe ATPase activity in the holothurian complex. However, this should not come as a surprise because stable multifunctional protein complexes from milk [9], placentae [10,11], eggs of sea urchins [12], and sea cucumbers [13,14] significantly differ in terms of protein composition. The protein composition of complexes from various sources may differ because of their varying biological functions, depending on the organisms and their biological fluids.

## 3. Discussion

As mentioned above, human placenta and milk, as well as sea urchin eggs, contain very stable multiprotein complexes with MWs of 1000 ± 100 kDa [9,10,11,12]. A very stable complex with an MW of 2000 kDa was identified in the extracts of sea cucumbers *Eupentacta fraudatrix* [13]. The very high stability of all these complexes indicates that their formation cannot result from a random association of different proteins [9,10,11,12,13]. Such very stable complexes containing many proteins and enzymes can have very extended biological properties compared to the individual components of the complexes. For example, the placenta complex possesses nine different enzymatic activities: DNase, RNase, ATPase, phosphatase, protease, amylase, catalase, peroxidase (H_2_O_2_-dependent), and oxidoreductase (H_2_O_2_-independent) activities [11]. All complexes contain metal ions, which may be important for forming contacts between the molecules of proteins and peptides. 

We analyzed the protease and DNase activities of the holothurian complex of 2000 kDa in this work. We found that the complex contains three types of proteases: metal-dependent, serine-like with trypsin substrate specificity, and thiol-like proteolytic enzymes. 

We found that the complex also contains several DNases, including metal-dependent and -independent enzymes. The maximum DNase activity of the complex was observed in the presence of Mn^2+^ ions, which is not typical for other DNases described in the literature with high activity in the presence of magnesium ions [29,30,31,32,33,34,35,36]. 

Many classical proteases [23,24,25] and DNases [29,30,31,32,33,34,35,36] from different organisms are dependent and independent on divalent metal ions. Divalent metal ions act as cofactors involved in chemical reactions catalyzed by some enzymes [17]. It is obvious that whole holothurian organisms also contain metal-dependent and -independent proteases, as well as nucleases, and some of them can be incorporated into a stable complex during molecular complex formation.

It should be assumed that the holothurian complex, like other complexes [9,10,11,12], can include many other enzymes. In addition, active sites with enzymatic activity can be formed at the junction of different proteins that do not exhibit catalytic activities in an individual state [9,10,11,12]. It cannot be ruled out that such complexes with extended enzymatic activities can play an important role in the vital activity of various organisms, including participation in the regeneration of cells and tissues. 

## 4. Materials and methods 

### 4.1. Chemicals 

High-purity reagents (SDS, Tris, different salts, bromophenol blue, EDTA, DTT, glycerol, urea, NH_4_HCO_3_, trifluoroacetic acid, and some other compounds) were obtained from Sigma (St. Louis, MO, USA). Histones H1, H2A, H2B, H3, and H4 of calf thymus were obtained from Sigma (product number 10223565001, St. Louis, MO, USA). Sepharose 4B was obtained from GE Healthcare Life Sciences (New York, NY, USA). Sea cucumber *Eupentacta fraudatrix* (holothurians) was collected from Peter the Great Bay in the Japan Sea. Samples of holothurians were frozen to −70 °C and stored until extract preparation.

### 4.2. Purification of Stable Complexes by Gel Filtration 

Ten holothurian homogenates were prepared as described in [13,14]. Homogenates were centrifuged at 1.7 × 10^3^× *g* for 50 min twice, filtered through an Amicon 0.1 nm filter, dialyzed four times against Milli-Q distilled water, and concentrated as described in [13,14]. The supernatant was additionally centrifuged at 1.2 × 10^3^× *g* at 4 °C for 15 min and subjected to FPLC gel filtration on Sepharose 4B to obtain a very stable protein complex (2000 kDa) as described in [13,14] (Appendix A). For analysis of the catalytic activities, the preparation of the resulting complex was subjected to repeated gel filtration on Sepharose 4B (Appendix A). 

### 4.3. Protease Activity of the Complex 

Protease activity of the holothurian complex was measured by the simple standard method using azocasein as a substrate [15]. The reaction mixture (300 μL) contained 3.3 mg/mL azocasein and 20 mM Tris-HCl (pH 7.5) or other buffers with pH in the range of 5.5 to 10.0 and 0.025 mg/mL protein complex. The mixture was incubated for 20–48 h at 37 °C, and then the reaction was stopped by adding 75 μL of 20% trichloroacetic acid before centrifugation at 1.2 × 10^3^× *g* for 2 min. An equal volume of 1.0 M NaOH was added to the mixture and incubated for 30 min at room temperature. Then, the mixture was centrifuged at 1.2 × 10^3^× *g* for 3 min, the supernatant was collected, and the absorbance was measured at a wavelength of 436 nm (A_436_) against the buffer (20 mM Tris-HCl (pH 7.5) or other buffers). The optical density was increased due to the elimination of azo dye from the azocasein. The specific activity was expressed as a change in A_436_.

pH dependencies were analyzed using different 50 mM buffers: MES-NaOH (pH 5.3–6.6), Tris-HCl (pH 6.0–8.6), and glycine-NaOH (pH 9.0–10.0). In some cases, MnCl_2_, MgCl_2_, ZnCl_2_, CaCl_2_, and CoCl_2_, (each at 0.1–7.0 mM) were added to the reaction mixtures.

To establish the type of protease activity of complex enzymes, the complex (0.1–0.2 mg/mL) was pre-incubated for 2 h at 25 °C with iodoacetamide (1–5 mM), PMSF (10 mM), or EDTA (10–100 mM), inhibiting selective thiol, serine, and metalloproteases, respectively. The aliquots of these mixtures (25–75 µL) were then added to the standard reaction mixture.

### 4.4. DNase Activity Assay

DNase activity of the complex was analyzed using supercoiled (sc) DNA pBluescript. The reaction mixture (20–40 µL) contained 50 mM Tris-HCl (pH 7.5), 2.5 mM MgCl_2_, 5.0 mM EDTA, 10 µg/mL (or 3.4 nM) sc DNA, and 0.05 mg/mL (or ~25 nM) protein complex. Reaction mixtures were incubated for 1–5 h at 37 °C. The cleavage products were detected using electrophoresis in 0.8% agarose gels; DNA was colored by ethidium bromide. The ethidium-bromide-stained gels were captured using a Sony DSC-F717 camera (Sony DSC-F717 camera; Sony Centre, Berlin, Germany). The hydrolysis of sc DNA results in the formation of its relaxed form with lower electrophoretic mobility. The initial native sc DNA always contains a small amount of hydrolyzed relaxed DNA. The relative intensity of DNA in different bands was analyzed by ImageQuant v5.2 (Molecular Dynamics). The complex activities were first determined as a decrease in the percent of sc DNA converted from the initial supercoiled to its relaxed form. The distribution of DNA between these two bands in the control (incubation of the DNA plasmid in the absence of the complex) was taken into account. 

pH dependencies were analyzed using different 50 mM buffers: MES-NaOH (pH 5.3–6.6), Tris-HCl (pH 6.0–8.6), and glycine-NaOH (pH 9.0–10.0). In some experiments, MnCl_2_, MgCl_2_, ZnCl_2_, CaCl_2_, and CoCl_2_, (each at 0.1–25.0 mM) were added to reaction mixtures.

### 4.5. In Situ Analysis of DNase Activity

Enzymes with DNase activity included in the composition of the complex were analyzed using the in situ method. A mixture containing 50 mM Tris-HCl (pH 7.5), 0.1% SDS, and 3 mg/mL complex was incubated for 1 h at 37 °C. Then, the holothurian complex was subjected to 4–18% SDS-PAGE using a gel containing calf thymus DNA (10 µg/mL). The gels were then washed with a solution of 4.0 M urea and twice with water to remove SDS and allow for protein renaturation. For polymeric DNA hydrolysis, the gel was incubated for 48 h at 37 °C in 20 mM Tris-HCl (pH 7.5) containing 1.0 mM EDTA, 2.0 mM CaCl_2_, and 5.0 mM MgCl_2_. The gel was then stained with 0.5% ethidium bromide to reveal the regions of DNA hydrolysis, which was detected as dark spots corresponding to an absence of stained DNA against the background of a fluorescent gel. The parallel longitudinal slices were used to detect the position of different proteins using standard Coomassie Blue staining. 

### 4.6. ATPase Activity of the Complex

For estimation of ATP-hydrolyzing activity, the reaction mixture (30 μL) contained 50 mM Tris-HCl, pH 7.5, 1 mM MgCl_2_, 0.3 mM EDTA, 1.0 mM ATP, and 0.05 mg/mL of holothurian complex, as described in [11]. The mixtures were incubated for 0–24 h at 37 °C, and 2 μL aliquots were applied to PEI cellulose plates. Thin-layer chromatography was carried out using 0.25 M potassium phosphate buffer with a pH of 7.0. The plates were then dried and photographed. The level of ATPase activity (%) was estimated based on the ratio of the relative fluorescence of non-hydrolyzed ATP and products of its hydrolysis estimated by ImageQuant 5.2. We found that the complex does not exhibit ATPase activity.

### 4.7. MALDI-TOF Mass Spectrometry Analysis

All mass spectra were obtained using a MALDI-TOF mass spectrometer (version 3.4; Bruker Daltonics, Bremen, Germany) with a nitrogen laser in positive reflector mode. The reaction mixtures for analysis of metal-independent proteases contained 50 mM Tris-HCl (pH 7.0), 0.8 mg mL, one of five histones (H1, H2A, H2B, H3, or H4), and 0.15 mg/mL (75 nM) of the complex. Before adding the preparations of the complex to the reaction mixture, they were dialyzed against 50 mM EDTA and then against three changes of Milli-Q water. The reaction mixtures were incubated with sea cucumber complex for analysis of the hydrolysis sites of five histones. After incubation of reaction mixtures for 0–20 h at 30 °C to 1 µL of the reaction mixture, 1 µL solution saturated with sinapinic acid (as a matrix) in a mixture of 0.2% trifluoroacetic acid and acetonitrile (1:2) was added. Then, 1 µL of the final mixtures were applied to MALDI standard steel plates, air-dried, and used for analysis.

Spectral analysis was performed in automatic mode (AutoXecute–automatic Run). The spectra were externally calibrated using standard mixtures of proteins (Protein Calibration Standard I, Bruker Daltonics). 

### 4.8. Statistical Analysis 

The results are presented as mean ± S.E. from at least two to three experiments for each analysis type using Microsoft Excel 2010. Analysis of peptide molecular masses corresponding to specific sites of histones splitting was performed using 7–8 spectra and Protein Calculator v3.3 (Scripps Research Institute; New York, NY, USA).

## 5. Conclusions

This article is the first to analyze protease and DNase enzymatic activities of sea cucumber *Eupentacta fraudatrix* (holothurian) very stable multiprotein complex. Our results demonstrate that this 2000 kDa complex possesses several protease and DNase activities. The complex metalloprotease is activated by several metal ions (Zn^2+^ > Mn^2+^ > Mg^2+^). The relative contribution of metalloproteases (~63.4%), serine-like protease (~30.5%), and thiol protease (~6.1%) to the total protease activity of the complex was estimated. The complex contains both metal-dependent and metal-independent DNases. Mg^2+^, Mn^2+^, and Co^2+^ ions strongly increased the DNase activity of the complex.

## Figures and Tables

**Figure 1 ijms-23-06677-f001:**
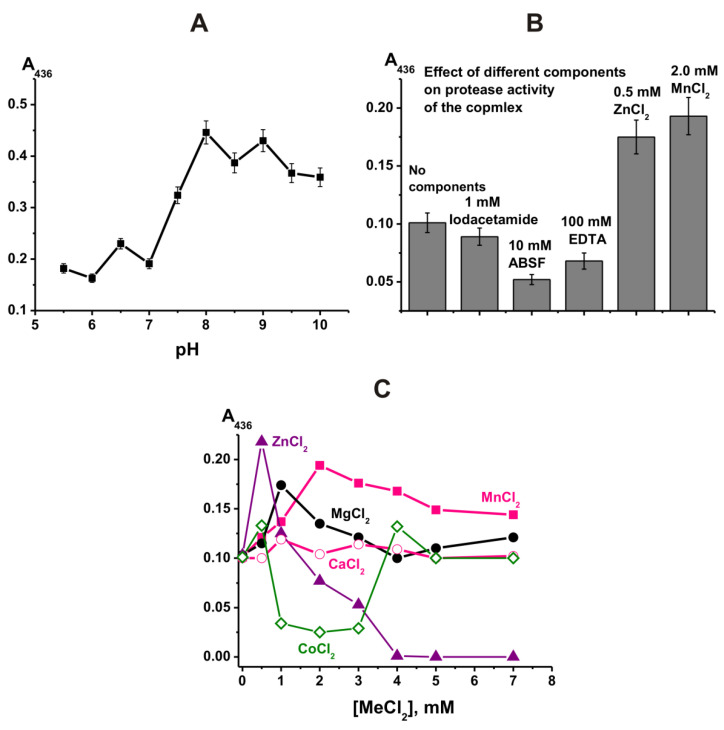
Dependence of the efficiency of azocasein (3.3 mg/mL) hydrolysis (A_436_) by a stable complex (0.025 mg/mL or ~1.25 × 10^−8^ M) on the pH of the reaction medium (**A**). Effect of preincubation of a stable complex with specific inhibitors of thiol (iodoacetamide), serine (ABSF), and metalloproteases (EDTA), as well as ZnCl_2_ and MnCl_2_, on its activity in the hydrolysis of azocasein (**B**). Dependence of the efficiency of azocasein hydrolysis on concentrations of different metal ions (**C**).

**Figure 2 ijms-23-06677-f002:**
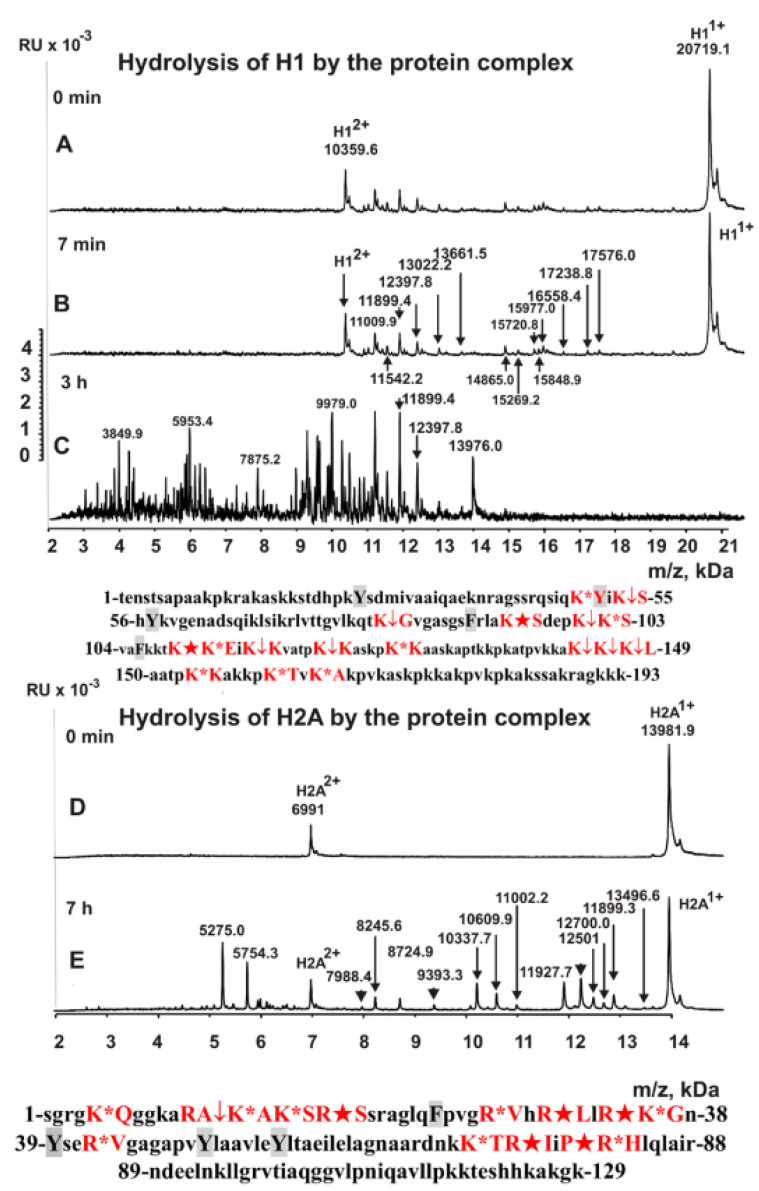
MALDI spectra correspond to 0.8 mg/mL H1 (**A**–**C**) and H2A histone (**D**,**E**) in the presence of sea cucumber stable complex (0.15 mg/mL or 7.5 × 10^−8^ M). Below the spectra, the protein sequences of histones H1 and H2A and the sites of their hydrolyses by the stable complex are shown. Major cleavage sites are indicated by large stars (★), moderate sites by arrows (↓), and minor sites by colons (:) (**A**–**D**). Aromatic residues of the protein sequence are marked in grey, after which no histone hydrolysis is observed. RU: relative values.

**Figure 3 ijms-23-06677-f003:**
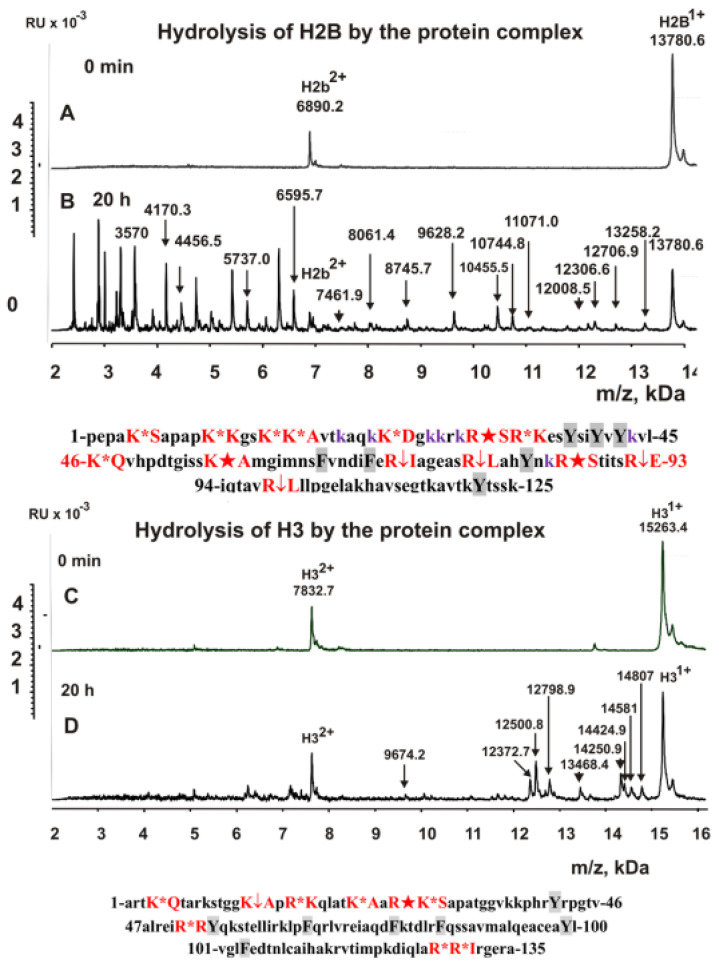
MALDI spectra correspond to 0.8 mg/mL H2B (**A**,**B**) and H3 histone (**C**,**D**) in the presence of sea cucumber stable complex (0.15 mg/mL or 7.5 × 10^−8^ M). Below the spectra, the protein sequences of histones H2B and H3 and the sites of their hydrolyses by the stable complex are shown. Major cleavage sites are indicated by large stars (★), moderate sites by arrows (↓), and minor sites by colons (:) (**A**–**D**). Aromatic residues of the protein sequence are marked in grey, after which no histone hydrolysis is observed. RU: relative values.

**Figure 4 ijms-23-06677-f004:**
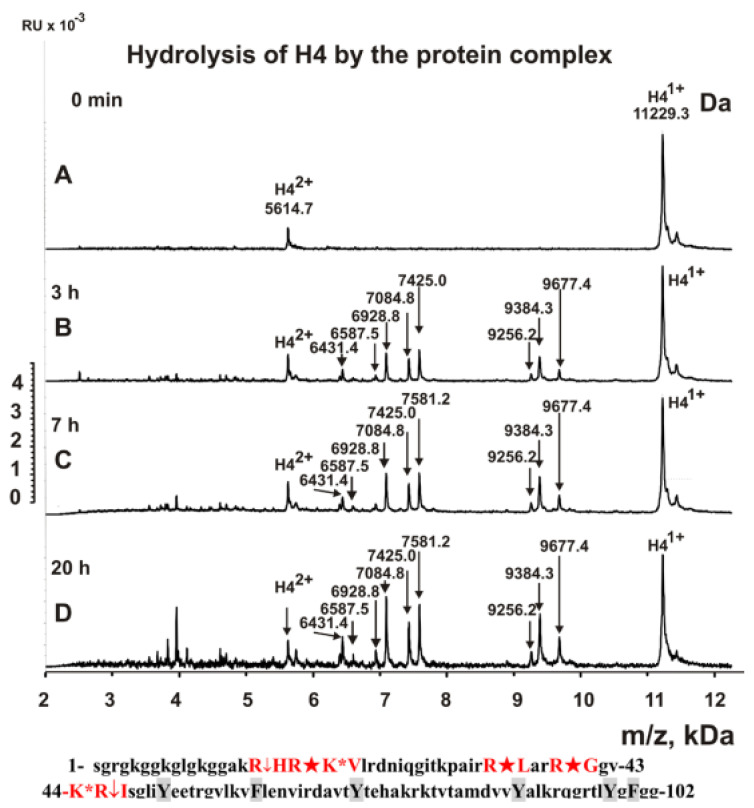
MALDI spectra correspond to H4 histone (0.8 mg/mL) over time hydrolysis in the presence of sea cucumber stable complex (0.15 mg/mL or 7.5 × 10^−8^ M). Major cleavage sites are indicated by large stars (★), moderate sites by arrows (↓), and minor sites by colons (:) (**A**–**D**). Below the spectra, the protein sequence of histone H4 and the sites of its hydrolysis by the stable complex are shown. Aromatic residues of the protein sequence are marked in grey, after which no histone hydrolysis is observed. RU: relative values.

**Figure 5 ijms-23-06677-f005:**
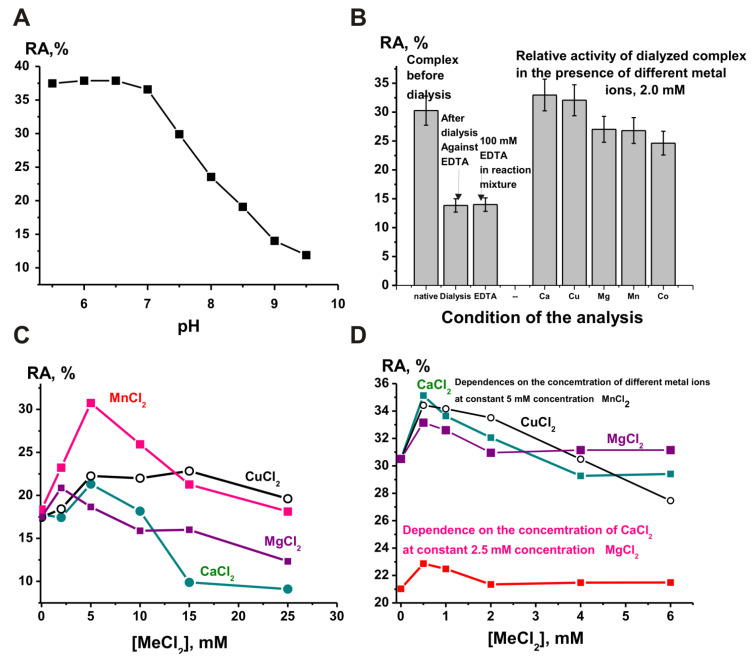
Dependence of the DNA efficiency of hydrolysis by a stable complex (0.015 mg/mL) on the pH of the reaction mixture (**A**). Effect of the complex preincubation with EDTA, addition of EDTA in reaction mixture, and different metal ions (2.0 mM) on the activity of a complex dialyzed against EDTA (**B**). Dependence of the efficiency of DNA hydrolysis by the complex dialyzed against EDTA on concentrations of different metal ions (**C**). Dependence of the DNase activity of the complex on the concentration of CaCl_2_ at a constant optimal concentration of MgCl_2_ (2.5 mM), as well as the concentration of chlorides of different metals (Ca, Cu, and Mg) at a fixed concentration (5.0 mM) of MnCl_2_ (**D**).

**Figure 6 ijms-23-06677-f006:**
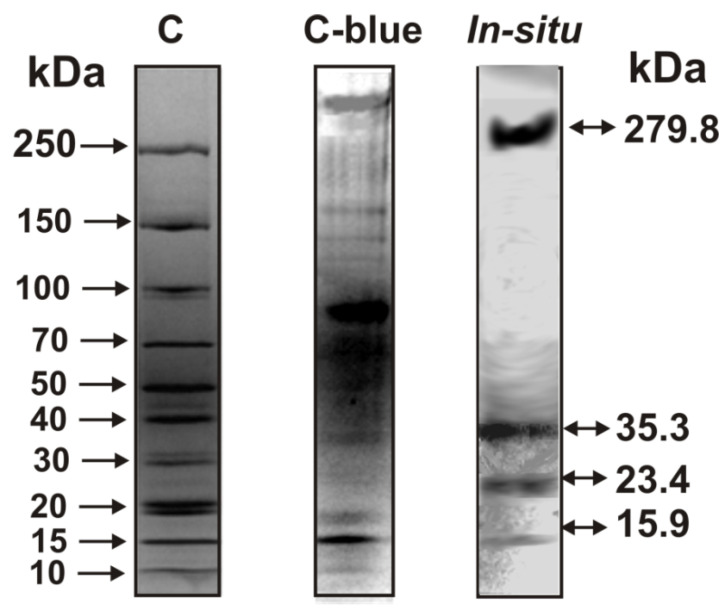
In situ assay of DNase activity of proteins of a stable complex (15 µg) using reducing conditions. DNase activity was revealed as dark bands on the fluorescent background (lane in situ) by ethidium bromide staining. A part of the gel corresponding to the destructed complex was stained with Coomassie R250 to show the position of complex proteins (lane C-blue). Lane C corresponds to proteins with known MWs. See Materials and Methods for further details.

## Data Availability

Data is contained within the article or Appendix A.

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
