# Peer review of "Protease and DNase Activities of a Very Stable High-Molecular-Mass Multiprotein Complex from Sea Cucumber Eupentacta fraudatrix"

_ijms, 2022, doi:10.3390/ijms23126677_

Round 1

Reviewer 1 Report

The abstract is too far to be descriptive of the work. It will be better to avoid preliminary considerations of organs and to directly focus on the molecular entities involved in their self-renewal. Please re-write

Very strange keywords!!!

Introduction:

“Common, 733 specific proteins, including 34 protein complexes, were found.” Maybe “included in …”

The introduction lacks of many recent references and should be improved. For example “ In connection with the expansion of the biological functions of stable complexes, it should be noted that they possess several metal-dependent and independent catalytic activities. In marine organisms capable of rapid regeneration, unique, highly stable complexes can exist” many examples are reported in the literature as:

DOI: 10.1038/s41598-018-22861-1

Results:

“by a complex of five histones using MALDI-TOF microscopy (in the absence of metal ions)” Why these five histones? How the metal ions were depleted? MALDI-TOF is not a microscopy

Discussion and Conclusions

These sections do not critically analyse data… Are the metal divalent ion series in agreement with the enzymatic activity of the same/similar multiprotein complexes? Which are chemical explanations?

Author Response

The abstract is too far to be descriptive of the work. It will be better to avoid preliminary considerations of organs and to directly focus on the molecular entities involved in their self-renewal. Please re-write

Answer:

Sorry, but the abstract first states why it is interesting and important to study the biological components of holothurians. Then the main data concerning the detection of protease and DNase activities in the stable complex are given. Taking into account your remark, it was added in the summary that metal-independent proteases of the complex hydrolyze proteins at trypsin-specific sites (after Lys and Arg).

Very strange keywords!!!

Answer:

It was corrected 

Introduction:

“Common, 733 specific proteins, including 34 protein complexes, were found.” Maybe “included in …”

Answer:

It was corrected

The introduction lacks of many recent references and should be improved. For example “ In connection with the expansion of the biological functions of stable complexes, it should be noted that they possess several metal-dependent and independent catalytic activities. In marine organisms capable of rapid regeneration, unique, highly stable complexes can exist” many examples are reported in the literature as:

DOI: 10.1038/s41598-018-22861-1

Answer:

It was included in the article

Results:

“by a complex of five histones using MALDI-TOF microscopy (in the absence of metal ions)” Why these five histones? How the metal ions were depleted? MALDI-TOF is not a microscopy

Answer:

It was corrected

Discussion and Conclusions

These sections do not critically analyse data… Are the metal divalent ion series in agreement with the enzymatic activity of the same/similar multiprotein complexes? Which are chemical explanations?

Answer:

Sorry, but it's not clear what to discuss in detail. Based on your comments, the following has been added

Many classical proteases [17-19] and DNases [23-28] from different organisms are dependent and independent on divalent metal ions. Divalent metal ions act as cofactors involved in chemical reactions catalyzed by some enzymes. Obviously that whole holothuria organisms also contain metal-dependent and independent proteases and nucleases and some of them can be incorporated into a stable complex during high-molecular complex formation.

Thank you for your helpful comments.

Best wishes

Professor George A. Nevinky

Reviewer 2 Report

The manuscript reports on the analysis of possible enzymatic activities of a stable protein complex related to sea cucumber Eupentacta fraudatrix.

The topic fits the journal aims.

The title correlates with the content of the article, but could be shortened (i.e., remove the “two mega Dalton”?).

The abstract reports a clear summary of the article findings, but could be better rephrased by providing an English editing.

All sections are required for a complete understanding.

Nevertheless, there are minor issues that requires to be addressed prior proceeding to acceptance for publication.

First of all, the introduction could benefit of additional references and mentioning a more complete overview, in order to emphasize the scientific soundness of the presented findings in terms of exchange, whether no NMR exchange data should be available, e.g. (Int. J. Pharm, 2018, 548, 474–479, Int. J. Mol. Sci. 2020, 21, 6804, J. Biomed. Mater. Res. A, 104A, 2016, 1668-1679) and many others.

Fig. 1 could be better presented: specifically, Fig. 1C, can be misunderstood by a broader reading prethora due to the presence of the “A” (capital letter) referring to the vertical axis: please, amend.

Fig. 6 reports on very undefined bands for “C-blue” and “In-situ”: it is suggested to improve the resolution/result in order to improve the results presentation.

The statistical programme used for analysis is missing in the method section.

The whole manuscript would strongly benefit an English editing (with an emphasis on the “Fundings” section”.

Check for typos.

Please, when presenting statistical significance, conform to the mostly used and accepted annotation: e.g., provide for stars (*,**,***). The manuscript will benefit remarkably and the readers plethora could be easily extended, thanks to a more easy understanding.

The conclusion section, although satisfactory, could be extended to better explain the overall findings: please amend the section title. Instead of “Conclusion” it is suggested to name it as “Conclusions”.

Author Response

Reviewer 2

The manuscript reports on the analysis of possible enzymatic activities of a stable protein complex related to sea cucumber Eupentacta fraudatrix.

The topic fits the journal aims.

The title correlates with the content of the article, but could be shortened (i.e., remove the “two mega Dalton”?).

Answer:

It was done

The abstract reports a clear summary of the article findings, but could be better rephrased by providing an English editing.

Answer:

It was done

All sections are required for a complete understanding. Nevertheless, there are minor issues that requires to be addressed prior proceeding to acceptance for publication

First of all, the introduction could benefit of additional references and mentioning a more complete overview, in order to emphasize the scientific soundness of the presented findings in terms of exchange, whether no NMR exchange data should be available, e.g. (Int. J. Pharm, 2018, 548, 474–479, Int. J. Mol. Sci. 2020, 21, 6804, J. Biomed. Mater. Res. A, 104A, 2016, 1668-1679) and many others.

Answer:

Answer:

We have included two of these articles in the introduction, but the content of the third is far from the topic of the article.

Fig. 1 could be better presented: specifically, Fig. 1C, can be misunderstood by a broader reading prethora due to the presence of the “A” (capital letter) referring to the vertical axis: please, amend.

Answer:

We have corrected the Figure 1.

Fig. 6 reports on very undefined bands for “C-blue” and “In-situ”: it is suggested to improve the resolution/result in order to improve the results presentation.

Answer:

Sorry, but depending on the number of different DNases and their relative activity, as well as the level of their glycosylation and the degree of destruction of the complex, the spots corresponding to DNA hydrolysis may be wider or narrower and the movement of free enzymes in the gel is sometimes difficult, since part of the complex remains intact in pockets of the gel, which interferes with the entry of free proteins into the gel. Taking this into account, it is a difficult task to obtain a beautiful picture as in the case of standard proteins with a known molecular weight that do not interact with each other in contrast to a strong complex. Of several new gels, one turned out to be more successful and we replaced the previous one with a new gel in Figure 6.

The statistical programme used for analysis is missing in the method section.

 Answer:

It was added

The whole manuscript would strongly benefit an English editing (with an emphasis on the “Fundings” section” 

Answer:

English has been tested

Please, when presenting statistical significance, conform to the mostly used and accepted annotation: e.g., provide for stars (*,**,***). The manuscript will benefit remarkably and the readers plethora could be easily extended, thanks to a more easy understanding.

Answer:

Sorry, but, from our point of view, this article does not contain any statistical comparisons, in which case it would be important to use (*,**,***).

The conclusion section, although satisfactory, could be extended to better explain the overall findings: please amend the section title. Instead of “Conclusion” it is suggested to name it as “Conclusions”.

Answer:

It was added

Thank you for your helpful comments.

Best wishes

Professor George A. Nevinky

Round 2

Reviewer 1 Report

Actual references 20 and 21 are the same. The authors should check for all references!!